

# Effects of various factors on Doppler flow ultrasonic radial and coccygeal artery systolic blood pressure measurements in privately-owned, conscious dogs

Allison P. Mooney[1], Dianne I. Mawby[1], Joshua M. Price[2] and Jacqueline C. Whittemore[1]

[1] Department of Small Animal Clinical Sciences, University of Tennessee—Knoxville, Knoxville, TN, United States
[2] Office of Information Technology, University of Tennessee—Knoxville, Knoxville, TN, United States

## ABSTRACT

**Objective**. The purpose of this study was to assess the effects of age, body condition score (BCS) and muscle condition score (MCS) on indirect radial and coccygeal Doppler systolic arterial blood pressure (SAP) measurements in dogs.

**Methods**. Sixty-two privately-owned dogs were enrolled between June and July 2016. The BCS and MCS were determined by two investigators. Blood pressure was measured per published guidelines and using headphones, and the order of measurement site was randomized. Dogs were positioned in right lateral recumbency for radial measurements and sternal recumbency or standing for coccygeal measurements. Associations between SAP and other variables were assessed by correlation coefficients and analysis of covariance.

**Results**. Radial and coccygeal SAP measurements were moderately correlated ($r = 0.45$, $P < 0.01$). Radial SAP measurements were higher than coccygeal SAP measurements (mean difference 9 mmHg, $P < 0.01$), but discordance occurred in both directions. No difference was observed between the first measurement taken, the average of measurements 2–6, or the average of all 6 measurements for either the radial (128, 129, and 129 mmHg; $P = 0.36$) or coccygeal (121, 122, and 122 mmHg; $P = 0.82$) site. Associations were not found between SAP measurements for either site and age, weight, BCS, MCS, anxiety score, or cuff size. Heart rate decreased significantly from the start of acclimation to the end of the first data collection series regardless of site ($P < 0.01$).

**Conclusions and Clinical Relevance**. Initial measurement site can be based on patient and operator preference given lack of associations with patient variables, but the same site should be used for serial SAP measurements given discordant results between sites.

Corresponding author
Jacqueline C. Whittemore,
jwhittemore@utk.edu

# INTRODUCTION

Due to the lack of overt clinical signs directly attributable to systemic hypertension, it is considered one of the most under-diagnosed systemic illnesses in companion animals

(*Acierno & Labato, 2004*; *Brown et al., 2007*). Prolonged systemic hypertension is associated with damage to the kidneys, eyes, brain, and heart (*Brown & Henik, 1998*; *Brown et al., 2007*; *Hsiang, Lien & Huang, 2008*; *Carr & Egner, 2009*). In dogs, systemic hypertension can occur secondary to diseases including kidney disease, hyperadrenocorticism, and diabetes mellitus (*Bodey & Michell, 1996*; *Brown & Henik, 1998*; *Brown et al., 2007*; *Hsiang, Lien & Huang, 2008*; *Carr & Egner, 2009*). Due the impact of hypertension on long-term outcome, routine surveillance for hypertension is recommended for dogs displaying signs consistent with end organ damage or that have been diagnosed with diseases or conditions associated with secondary hypertension (*Acierno & Labato, 2004*; *Henik, Dolson & Wenholz, 2005*; *Brown et al., 2007*). While there is some debate regarding the relationship between advancing age and hypertension, conditions that can cause secondary hypertension are more often observed in geriatric pets, so this population should be monitored for the development of those diseases (*Brown et al., 2007*).

Hypotension requiring vasopressor therapy is associated with a 3.4-fold increase in the risk of death in dogs with sepsis (*Kenney et al., 2010*). Hypotension at admission has similarly been associated with decreased survival in critically-ill cats (*Simpson et al., 2007*). Silverstein, et al. further demonstrated that an increase in systolic Doppler blood pressure of $\geq$20 mmHg increased the chance of survival to discharge in hypotensive cats (*Silverstein et al., 2008*) and that dogs that had normalization of hypotension within the first hour of fluid resuscitation were more likely to survive to discharge (*Silverstein, Kleiner & Drobatz, 2012*). Thus, accurate determination of indirect blood pressure measurement is key to identify patients with both hyper- and hypotension in order to optimize outcome.

Although direct arterial measurement of blood pressure remains the gold standard, the pain and specialized equipment associated with this method limits its use in a clinical setting (*Sawyer, Guikema & Siegel, 2004*; *Sawyer et al., 1991*; *Gains et al., 1995*; *Bodey et al., 1996*). Indirect methods of measuring blood pressure are less invasive and technically demanding and are, therefore, more practical for routine monitoring of blood pressure in conscious dogs. Currently, the two most frequently used modalities of indirect blood pressure measurement are ultrasonic Doppler flow monitors and automated oscillometric devices. The Doppler flow method has been shown to be more efficient and to generate more precise measurements of systolic arterial blood pressure (SAP) in conscious dogs in a clinical setting (*Hsiang, Lien & Huang, 2008*; *Wernick et al., 2012*). Furthermore, oscillometry can underestimate increased SAP and overestimate decreased SAP, resulting in failure to diagnose hyper- and hypotension, respectively (*Sawyer, Guikema & Siegel, 2004*; *Sawyer et al., 1991*; *Gains et al., 1995*; *Bodey et al., 1996*; *Wernick et al., 2012*; *Vachon, Belanger & Burns, 2014*). In addition to the variability that exists between the two methods described above, both the location of cuff placement, as well as the body position of the animal, can significantly affect the accuracy of indirect blood pressure measurements (*Bodey et al., 1994*; *Bodey et al., 1996*; *Rondeau, Mackalonis & Hess, 2013*; *Scansen et al., 2014*). Selection of an appropriately-sized blood pressure cuff also impacts the accuracy of indirect blood pressure measurements, with cuff-size inversely correlated with blood pressure measurement results (*Valtonen & Eriksson, 1970*; *Bodey et al., 1994*; *Sparkes et al., 1999*). Excitement or anxiety experienced during the measurement process, a phenomenon

known as white-coat hypertension, can also result in erroneous measurement results (*Belew, Barlett & Brown, 1999*; *Brown et al., 2007*; *Bragg et al., 2015*). Finally, patient-specific factors can affect the accuracy of indirect blood pressure measurements. Although no association has been identified between body condition score (BCS) or weight and indirect blood pressure measurement results in either cats (*Sparkes et al., 1999*) or dogs (*Remillard, Ross & Eddy, 1991*; *Bosiack et al., 2010*), *Whittemore, Nystrom & Mawby (2017)* recently found that muscle mass is inversely associated with SAP measurements taken using the radial, but not the coccygeal, artery in cats. Prior studies assessing the effects of weight and BCS on SAP measurements did not separately assess muscle condition score (MCS). To the authors' knowledge, the separate effects of BCS and MCS on correlations between indirect radial and coccygeal SAP measurements in dogs are unknown.

The purpose of this study was to assess the impacts of BCS and MCS on correlations between indirect radial and coccygeal SAP measurements taken using the Doppler method in privately-owned dogs. Secondary objectives of the study were to assess for associations between SAP measurement results and anxiety score, cuff size, and heart rate and to compare time required for collection of a complete series of readings at each site.

## MATERIALS & METHODS

### Study population

This study was conducted at the University of Tennessee's Veterinary Medical Center and was approved by the Institutional Animal Care and Use Committee of the University of Tennessee, Knoxville (protocol number 2426).

Between June and July 2016, privately-owned dogs were enrolled in the study based on initial BCS determination assigned by one investigator (APM) using a 5-point scale (*Baldwin et al., 2010*). An enrollment goal of 20 dogs each per whole integer BCS was established for a total target sample size of 100 dogs. After 20 dogs had been enrolled within a whole integer score, no additional dogs of the same BCS were enrolled.

Privately-owned dogs from the veterinary teaching hospital population as well as those belonging to faculty, staff, and students were recruited. Informed consent was obtained for each dog prior to enrollment. Prior to data collection, owners were asked to provide information regarding any known medical conditions and/or current medications. Dogs that did not have a tail, had been anesthetized within the previous 12 hrs, or became fractious or intolerant of manipulation were excluded from the study. Participants diagnosed with local or systemic diseases were not excluded, nor were those dogs receiving medications, consistent with previous studies (*King et al., 2001*; *Hsiang, Lien & Huang, 2008*; *Bosiack et al., 2010*; *Wernick et al., 2012*). Immediately upon entry into the exam room, both the weight and initial heart rate were recorded.

### Determination of BCS and MCS

All investigators that completed BCS and MCS scoring received training in BCS and MCS assignment from a board-certified veterinary nutritionist prior to the start of data collection. Investigators independently assigned a BCS and MCS to each participant using a 5-point scale and a 4-point scale, respectively (*Baldwin et al., 2010*; *Michel et al., 2011*).

The initial BCS used for sample stratification was assigned to each dog by one investigator (APM). Body condition scores and MCS from two additional investigators (DIM, JCW) were used for statistical analyses. These investigators were blinded to BCS and MCS scores for the other investigators, as well as all SAP measurement results, until completion of sample collection for the study.

### Randomization

Using a randomized number generator (https://www.random.org, accessed May 15 2016), a randomization table was generated to determine the initial site of SAP measurement (radial versus coccygeal artery) for the target number of participants for each BCS category. Based on the initial BCS assigned, dogs were allocated into measurement groups using the next available assignment in the randomization table. If a participant was excluded from the study after assignment in the randomization table, its name was crossed out once on the table and that randomization assignment was recycled for the next participant.

### Blood pressure measurement

Blood pressure measurements were obtained following the guidelines provided in the American College of Veterinary Internal Medicine Consensus Statement (*Brown et al., 2007*). Briefly, both the dog and owner (if possible) were brought into a quiet exam room for data collection. If the owner was unable to be present for both pressure series, then either the investigator collected all measurements without assistance or a veterinary assistant assisted with the entire data collection series. The circumference of both the left mid-antebrachium and the base of the tail were measured using a soft measuring tape, and cuff size was selected to be approximately 40% of the circumference of each appendage. The location where the Doppler crystal was to be placed for the first pressure series was shaved—on the palmar surface of the left front foot between the carpal and metacarpal pads or the ventral surface of the tail just distal to site of cuff placement. The dog was then allowed to acclimate for 10 min to the environment and measurement personnel. Heart rates at the beginning and end of acclimation were recorded, as well as 'start' and 'stop' times for the acclimation period.

After acclimation was complete, the blood pressure cuff was applied to the first measurement site as per group assignment, either just proximal to the carpus on the left mid-antebrachium or just distal to the base of the tail. For radial SAP measurements, dogs were placed in right lateral recumbency with the left forelimb positioned so that the cuff was at the same level as the heart. If the left antebrachium was unavailable for measurements (due to injury, IV catheter placement, etc.), the animal was placed in left lateral recumbency and the right antebrachium was used. For coccygeal SAP measurements, the dog was allowed to either stand or lay in sternal recumbency. Ultrasonic coupling gel was applied to the concave side of a flat infant Doppler probe, which was then positioned perpendicular over the artery. Using the same sphygmomanometer (Riester Ri-san® aneroid sphygmomanometer; Riestar Direct, Ventura, CA, USA) and Doppler unit with probe (Model 811-B Doppler ultrasonic Flow Detector with flat infant probe; Parks Medical Electronics, Inc., Aloha, OR, USA) for every SAP measurement, the cuff was inflated to

approximately 20 mmHg above the point at which blood flow was no longer audible. Air was slowly and completely released from the cuff, and the pressure at which flow was first heard was recorded as the SAP. The process was repeated until 6 consistent readings were obtained. Heart rates at the beginning and end of each pressure series were recorded using Doppler ultrasonic pulse detection. Headphones were worn by the investigator for the entire duration that the Doppler was turned on to limit the potential of noise from the machine to contribute to 'white coat' effects. An anxiety score (ranging from 0 to 3) was assigned to the participant after completion of data collection (*Scansen et al., 2014*), with a calm dog requiring no restraint being assigned a 0 and a highly anxious dog requiring active restraint assigned a 3. As noted above, dogs that were refractory to SAP measurement at either site were excluded from the study. After completion of data collection at the first site, instrumentation was removed and the alternate site was clipped. Then the measurement procedure was repeated for the alternate site starting with the 10-minute acclimation period.

## Data entry

After data collection was completed on a subject, as described above, the data collection sheet was turned in to a person unrelated to study. That person was responsible for entry of all results into the study database, which was not available to any of the investigators until after completion of data collection for all subjects in the study.

## Statistical analyses

Descriptive statistics were generated for each parameter. Continuous measures were analyzed for normality using the Shapiro–Wilk test and for the presence of outliers using the box-and-whisker plots. Equality of variances was analyzed with Levene's test for equality of variances. Parameters with normally distributed data were reported as mean $\pm$ standard deviation, with non-parametrically distributed data reported as median (range). Interclass correlation coefficients (weighted κ) were calculated for inter-rater reliability for BCS and MCS scores assigned by two investigators (DIM, JCW), after which the mean of the 2 scores was used for all further analyses. Mean SAP measurements greater than 150 mmHg were considered consistent with hypertension, based on previous guidelines (*Brown et al., 2007*). Mean SAP measurements less than 90 mmHg were considered consistent with hypotension (*Ateca, Dombrowski & Silverstein, 2015*). Repeated measures analysis of variance (ANOVA) was used to assess for differences in heart rate at the start and conclusion of each measurement series, as well as between the initial SAP reading, the mean of blood pressure measurements 2–6 (BP 2–6), and the mean of all 6 readings for each site. In keeping with ACVIM consensus guidelines (*Brown et al., 2007*), the mean of BP 2–6 was used for all remaining analyses. Finally, a paired sample student's t test was used to compare time required for completing data collection at each site.

Pearson product moment correlation coefficients were calculated to assess correlations between the average of BP 2–6 for the radial and coccygeal sites, as well as between the average of BP 2–6 for each site and possible covariates (age, weight, heart rate, BCS, MCS, and limb or tail circumference as appropriate). Regression analysis was used to determine

**Table 1  Baseline demographics for 62 privately-owned conscious dogs in which indirect radial and coccygeal systolic arterial blood pressure (SAP) measurements were collected.** Values are reported as median (range).

| | Enrollment BCS | | | | | |
|---|---|---|---|---|---|---|
| Variable | 1 (n = 2) | 2 (n = 6) | 3 (n = 20) | 4 (n = 20) | 5 (n = 14) | All dogs (n = 62) |
| Age (years) | 13.5 (12–15) | 10 (3–11) | 7 (1–13) | 7 (3–14) | 9.5 (5–14) | 8 (1–15) |
| Gender | 2 MC | 1 FS, 5 MC | 10 FS, 11 MC | 8 FS, 11 MC | 8 FS, 6 MC | 27 FS, 35 MC |
| Weight (kg) | 22.7 (13.3–31) | 7.1 (2.7–41.4) | 20.9 (2.7–54.3) | 16.2 (2.3–50.4) | 14.1 (4.9–61.3) | 18.6 (2.1–61.3) |
| Anxiety score | 1 (NA) | 1 (1–3) | 1 (0–2) | 1 (0–2) | 1 (0–3) | 1 (0–3) |
| Mean BCS[a] | 1.25 (1–1.5) | 2 (1.5–2) | 3 (2–3.5) | 4 (3–5) | 5 (4.5–5) | 3.5 (1.0–5.0) |
| Mean MCS[a] | 0.25 (0–0.5) | 2 (1–3) | 3 (1–3) | 3 (1–3) | 2.5 (1.5–3) | 3.0 (0.0–3.0) |
| Radial SAP (mmHg) | 120 ± 5.2 | 139 ± 18.8 | 126 ± 20.5 | 126 ± 16.4 | 136 ± 15.4 | 130 ± 18.0 |
| Coccygeal SAP (mmHg) | 110 ± 29.6 | 129 ± 17.5 | 117 ± 17.5 | 116 ± 19.0 | 129 ± 21.2 | 120 ± 18.8 |

**Notes.**

BCS, body condition score; MCS, muscle condition score; MC, male castrated; FS, female spayed.

[a]The mean of scores for 2 investigators (JCW, DIM) were used for statistical analyses.

variance inflation factors to assess for collinearity between possible covariates (age, weight, the mean of starting and final heart rate for each pressure series, BCS, MCS, anxiety score, and cuff size as a percent of appendage circumference).

A mixed effects crossover design and corresponding ANCOVA was performed to determine if mean BP 2–6 differed between radial and coccygeal measurement sites and to assess whether the washout period between treatment sites was adequate. Period and Site were included as fixed effects. Age, weight, average heart rate, BCS, MCS, anxiety score, time required to complete the measurement series, and cuff size as a percentage of appendage circumference were initially included as covariates in the analysis. Dog nested within sequence was included as a random effect. A compound symmetry variance/covariance structure was incorporated into the model to account for constant covariates. Backwards variable selection was performed on the full model to determine which covariates explain significant variability in mean BP 2–6. Factors included in the final model were period, site, and average heart rate. The Shapiro–Wilk test of normality of the residuals was evaluated to ensure the assumptions of the statistical method had been met. Commercial statistical software packages MedCalc (MedCalc 15.8; MedCalc Software, Ostend, Belgium), SAS (SAS 9.4 release TS1M3; SAS Institute Inc., Cary, NC, USA) and SPSS (IBM SPSS Statistics for Windows, version 24; IBM Corp., Armonk, NY, USA) were used for all analyses. $P < 0.05$ was considered significant.

## RESULTS

Seventy dogs were enrolled in the study, of which eight were excluded due to intolerance of manipulation or SAP measurement at either the radial or coccygeal artery (n = 5) or unavailability of one of two investigators (JCW, DIM) to assign a BCS/MCS prior to subject discharge (n = 3). Initial BCS categorization for the 62 dogs that completed the study was BCS 1: two dogs, BCS 2: six dogs, BCS 3: 20 dogs, BCS 4: 20 dogs, and BCS 5: 14 dogs. Age, sex distribution, weight, anxiety score, mean BCS and MCS for measurements assigned by DIM and JCW, and mean radial and coccygeal SAP measurements are presented in Table 1.

Mixed breed dogs comprised the majority of the study population ($n = 24$), with three or less each of 29 breeds. The complete list of breeds and the number of dogs representative of those breeds are provided in Table S1. Of the 62 dogs that completed the study, 33 dogs were healthy and 29 had one or more known disease processes or conditions (see Data S1 for individual diagnoses). The interclass correlation coefficients for inter-rater reliability for BCS and MCS as assigned by DIM and JCW were $\kappa = 0.76$ ($P < 0.01$) and $\kappa = 0.57$ ($P < 0.01$), respectively.

As noted above, heart rate was determined at 4 time points throughout each measurement procedure (start of acclimation, end of acclimation, start of SAP measurements, end of SAP measurements). Mean heart rate differed significantly over time ($P < 0.01$) at each site. For the radial site, heart rate values ranged from 104 bpm (range: 68–180) at the beginning of the acclimation period to 93 bpm (range: 54–162) at the end of the procedure. Post-hoc tests indicated that radial heart rate values were significantly different at the beginning of the acclimation period when compared to heart rates at the three other time points ($P < 0.01$). For the coccygeal site, heart rate ranged from 104 bpm (range: 60–180) at the beginning of the acclimation period to 93 bpm (range: 60–162) at the end of the measurement period. Although there was a significant difference in heart rate during the acclimation period ($P < 0.01$), this difference was not evident across all time points based on post-hoc analysis. Interestingly, there was no significant difference between the first blood pressure taken, the average of BP 2–6, or the average of all six measurements for either the radial (128, 129, and 129 mmHg, respectively; $P = 0.36$) or coccygeal (121, 122, and 122 mmHg, respectively; $P = 0.82$) site. There was also no difference in the time required for collection of the complete SAP measurement series for the two sites (5 min for the radial site *vs* 4.6 min for the coccygeal site; $P = 0.30$).

One dog had a history of severe degenerative joint disease in the lower spine and tail and was markedly distressed by movements near, or manipulation of, the hind end. This dog had marked discordance in SAP readings (radial, 113 mmHg *vs* coccygeal, 224 mmHg) and was censored from the analyses of BP 2–6 results because collection of coccygeal measurements would not have been performed in this case given prior knowledge of its orthopedic disease. Even after censoring results for the previously described dog, radial and coccygeal artery blood pressure measurements were poorly correlated ($r = 0.45$, $P < 0.01$, Fig. 1). Passing-Bablok regression and Bland-Altman analyses did not reveal either constant or proportional bias.

Discordance in radial and coccygeal artery SAP measurements occurred in both directions across all BCS (Fig. 2) and MCS. There were no significant correlations between mean BP 2–6 for either site and any of the assessed possible covariates. Further, variance inflation factors were within the normal range for age, weight, the mean of starting and final heart rates for each SAP measurement series, BCS, MCS, anxiety score, and cuff size as a percent of appendage circumference. Based on lack of evidence of collinearity, all were included in the initial mixed effects crossover design. Age, weight, BCS, MCS, anxiety score, and cuff size as a percent of appendage circumference were not correlated with BP2–6 ($P > 0.2$). Thus, the final crossover model included measurement period, site, and mean heart rate. Of these, there was no significant effect of measurement period ($P > 0.2$)

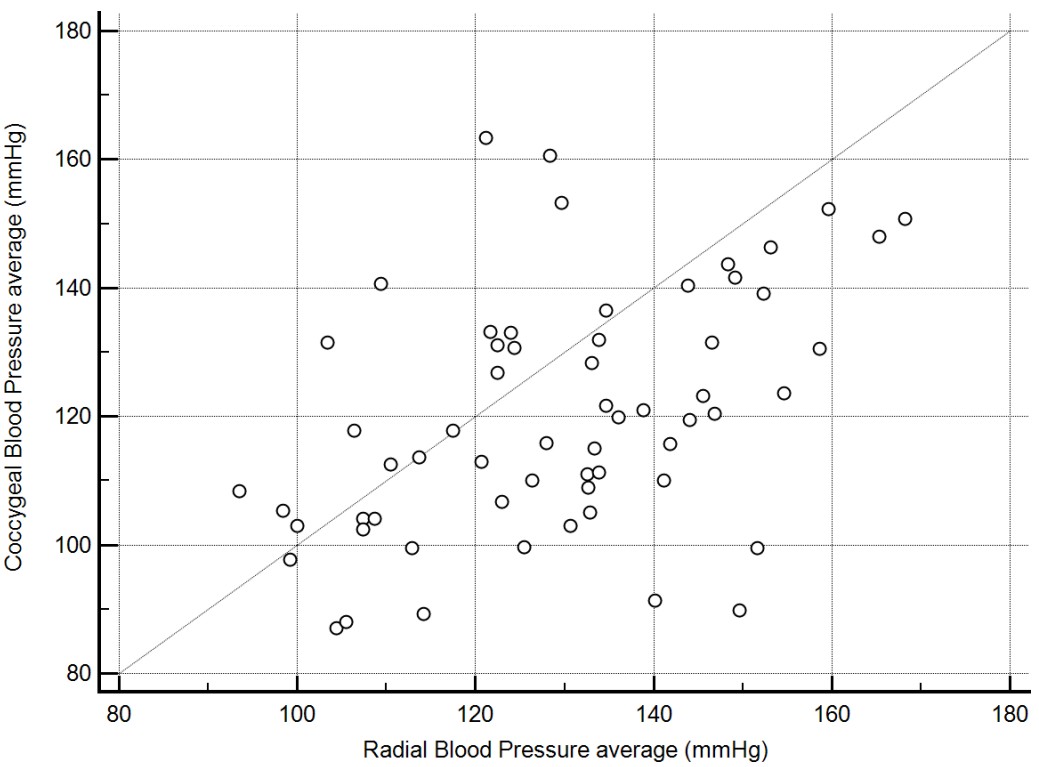

**Figure 1** Correlation between mean indirect radial and coccygeal systolic arterial blood pressure measurements collected via Doppler ultrasonic flow detector for 62 privately-owned conscious dogs.

on BP 2–6 results, thereby indicating that the washout period between treatments was sufficient. Radial artery BP 2–6 results significantly differed from coccygeal results ($F$-value 18.9, $P < 0.01$). Mean radial BP 2–6 was higher than coccygeal BP 2–6 (mean difference 9 mmHg, $P < 0.01$), but discordance occurred in both directions. A significant positive association was also found between BP 2–6 and mean heart rate ($F$-value 5.9, $P = 0.02$).

Mean radial and coccygeal BP 2–6 were $129 \pm 18.0$ mmHg and $120 \pm 18.8$ mmHg, respectively. Of the 62 dogs, 8 (12.9%) were categorized as hypertensive based on radial SAP measurements, vs 6 (9.7%) dogs based on coccygeal SAP measurements. Two dogs with histories of refractory hypertension (>190 mmHg) that were enrolled in the study were normotensive (mean 140 mmHg). No dogs were categorized as hypotensive based on radial SAP measurements *vs.* 2 (3.2%) dogs based on coccygeal SAP measurements.

## DISCUSSION

Radial and coccygeal artery blood pressure measurements were poorly correlated in this study. Although discordance between measurements taken at these two sites occurred in both directions, mean radial BP2–6 measurements were significantly higher than those obtained at the coccygeal artery. Consistent with previous reports (*Remillard, Ross & Eddy, 1991*; *Bosiack et al., 2010*), no significant associations were found between BP 2–6 at either site and age, BCS, or weight. Furthermore, no association was found between MCS and BP 2–6, in contrast to results from a recent study in cats (*Whittemore, Nystrom & Mawby,*

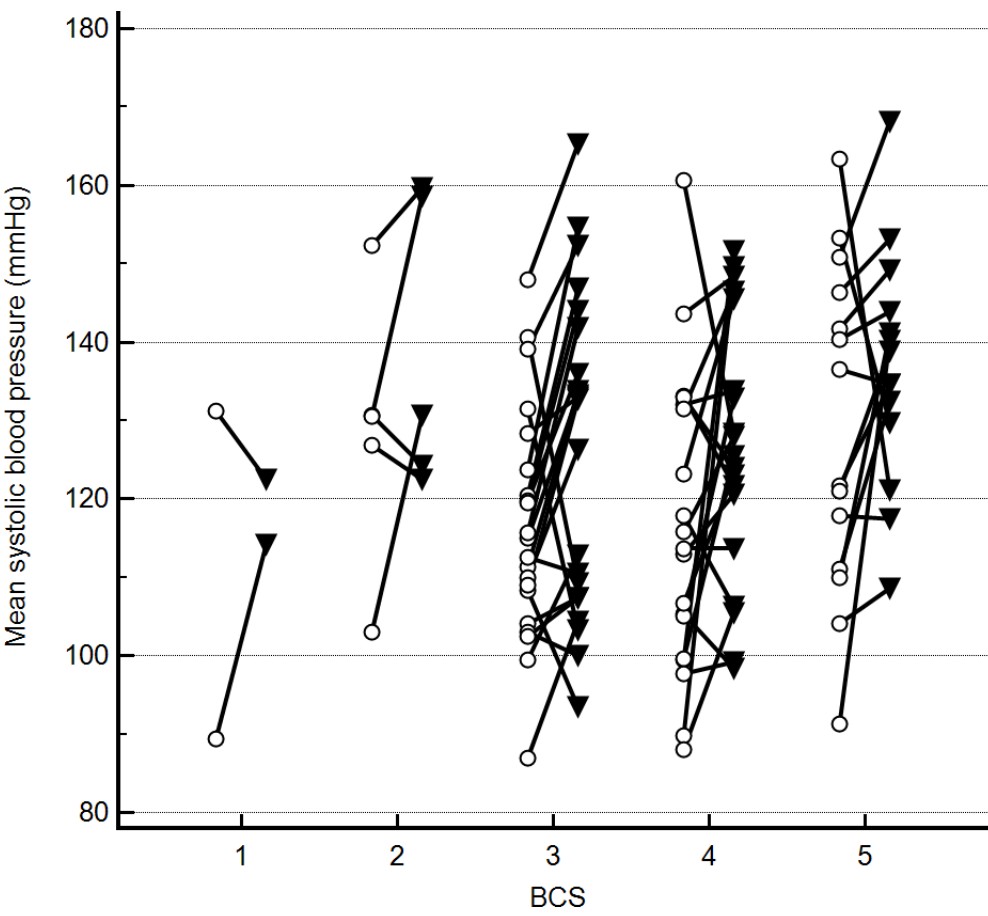

**Figure 2** **Mean indirect radial and coccygeal systolic arterial blood pressure measurements collected via Doppler ultrasonic flow detector in 62 privately-owned conscious dogs.** Dogs are grouped by enrollment body condition score (BCS) assignment. The open circles represent measurements taken using the coccygeal artery, whereas closed triangles represent measurements taken using the radial artery.

*2017*). Importantly, this finding indicates that the presence of sarcopenia need not affect site selection for indirect SAP measurements taking using the Doppler flow method in dogs. Thus, individual patient factors can be prioritized when selecting a blood pressure measurement site, the importance of which is underscored by the marked discordance in results in one dog with severe degenerative joint disease in this study.

Several previous studies have also identified discordance between indirect and direct blood pressure measurements taken at different sites (*Bodey & Michell, 1996*; *Rondeau, Mackalonis & Hess, 2013*; *Acierno et al., 2015*). *Bodey & Michell (1996)* found oscillometric systolic blood pressure measurements were significantly lower at the coccygeal artery compared to the radial artery in standing dogs. In contrast, the opposite was found for oscillometric blood pressure measurements taken in lateral recumbency in the same study (*Bodey & Michell, 1996*). The disagreement between our results and those obtained by Bodey and Michell in laterally recumbent dogs could be attributed to the differences in body position and the wide variety of dog breeds and conformations sampled in our study. Instead of requiring dogs to assume a standard position for tail measurements, participants
were allowed to either remain standing or lie in sternal recumbency. It is possible that the difference in the level of the coccygeal artery relative to the heart was greater in larger, deep-chested dogs allowed to remain standing than in smaller dogs of more moderate to shallow-chested conformation. This could have led to a greater discrepancy in results than would have been observed had the dogs all been positioned in sternal recumbency for coccygeal readings. The impact of body position on indirect blood pressure measurement is further highlighted by the results of Rondeau, Mackalonis, and Hess, in which the mean SAP in conscious dogs measured on the forelimb was significantly higher in the sitting position than in lateral recumbency (*Rondeau, Mackalonis & Hess, 2013*). Finally, significant differences have been noted in direct SAP measurements collected at different anatomic sites, particularly at the carpus and hindlimb (*Acierno et al., 2015*), suggesting an anatomic basis for discordance in pressures at different sites. Ultimately, the results of these studies underscore the importance of consistency in both measurement site and body positioning when obtaining indirect blood pressure readings, especially when monitoring for the progression of disease.

White-coat hypertension refers to an increase in blood pressure secondary to the stress and anxiety associated with the measurement process (*Belew, Barlett & Brown, 1999*; *Brown et al., 2007*; *Bragg et al., 2015*). It is possible that decreased tolerance of radial compared to coccygeal measurement resulted in a site-associated white coat effect, although there was no difference in anxiety scores or apparent tolerance of the procedure between the two sites. While no correlation was observed between anxiety score and SAP measurement or heart rate for either site, previous studies in human medicine have revealed that suppression of anger in both men (*Mills & Dimsdale, 1993*) and women (*Thomas, 1997*) can result in elevated blood pressure. It is possible that dogs that internalize stress, resulting in outwardly calm demeanors, might be more affected by white coat hypertension than overtly anxious dogs. Resting heart rates taken in hospital have been found to be significantly higher than measurements taken in the home environment for both cats (*Belew, Barlett & Brown, 1999*) and dogs (*Remillard, Ross & Eddy, 1991*; *Kallet, Cowgill & Kass, 1997*; *Bragg et al., 2015*). Although some studies observed concurrent increases in blood pressure and heart rate in a clinic setting (*Belew, Barlett & Brown, 1999*; *Bragg et al., 2015*), at least one study found no correlation between increases in heart rate and blood pressure (*Remillard, Ross & Eddy, 1991*). Although our study did not compare the heart rates at home with those obtained upon entry into the hospital, heart rates taken before and after acclimation and SAP measurement were compared. Heart rates declined significantly during data collection, and heart rates at the conclusion of SAP measurement at the radial site were significantly lower than heart rates at the end of acclimation or the beginning of data collection. Importantly, there was no significant difference between the first SAP reading taken in each dog at either site *vs.* the mean of the following five in spite of continuing changes in heart rate. This suggests that the positive association between average heart rate and mean SAP at each measurement site is more likely due to underlying physiologic processes *vs.* persistent white-coat effects.

Given caseload and personnel constraints in many practices, it can be challenging to consistently make time for adequate acclimation, as well as collection of a full series of SAP

measurements. The importance of measuring SAP prior to other interventions and allowing patients to acclimate to their surroundings—including veterinary personnel—prior to blood pressure measurement cannot be overstated, however. In one study comparing blood pressure measurement results for healthy cats and cats with experimentally-induced chronic kidney disease (*Belew, Barlett & Brown, 1999*), white-coat associated blood pressure increases were more marked and less quick to normalize in cats with kidney disease. Because routine blood pressure measurement is recommended for animals diagnosed with diseases associated with secondary hypertension (*Acierno & Labato, 2004*; *Henik, Dolson & Wenholz, 2005*; *Brown et al., 2007*) and they often are subjected to more frequent veterinary visits for diagnostic sampling and medical procedures, such patients are particularly vulnerable to being misdiagnosed with hypertension if not allowed adequate acclimation and/or if other procedures are performed prior to blood pressure measurement. Although the ACVIM consensus statement (*Brown et al., 2007*) recommends collecting a total of 3–7 measurements and discarding the first before determining the average SAP, there was no clinical or statistical difference between the first measurement, average of BP 2–6, or BP 1–6 in this study. This suggests that a single SAP measurement could be used in dogs with limited concern for reproducibility of the results, assuming that adequate acclimation of the subject occurs. In contrast, failure to perform an adequate acclimation cannot be overcome by collection of additional readings—as demonstrated by the two subjects in this study with previously diagnosed, apparently refractory hypertension that were repeatedly normotensive, both during and after the study, when acclimated for 10 min with technical personnel present. For these reasons, in cases where it is not feasible to do both, acclimation of the patient with personnel should be prioritized over collection of multiple readings in dogs.

Current guidelines for indirect measurement of blood pressure in dogs recommend selecting a cuff size that is approximately 40% of the appendage circumference (*Brown et al., 2007*). Undersized or oversized blood pressure cuffs have been shown to falsely elevate or lower indirect blood pressure measurements, respectively (*Valtonen & Eriksson, 1970*; *Bodey et al., 1994*; *Sparkes et al., 1999*). Although appendage circumference was measured in this study to allow selection of the ideal cuff size, actual percent circumference of the cuff ranged from 30 to 66% due to lack of availability of half-sizes. In spite of this, no association was found between cuff size as a percentage of circumference and SAP results at either site. This suggests that use of a suboptimal percentage due to cuff size limitations should have minimal clinical impact on blood pressure measurement.

One limitation of this study was the lack of direct arterial measurements to compare to indirect SAP measurement results. However, as discussed above, direct SAP measurement is invasive and can be painful, limiting its utility in evaluation of blood pressure in conscious animals on an outpatient basis. Additionally, direct SAP results vary depending on the vessel used and position of the animal, as demonstrated by *Acierno et al. (2015)*. Thus, for comparison to direct SAP results to be valid, direct and indirect measurements would need to be taken at both the radial and coccygeal sites. Because the coccygeal artery is not a paired artery, simultaneous collection of direct arterial and indirect SAP measurements would not be possible. To avoid confounding impacts of arterial catheterization on local

vascular tone, direct and indirect SAP measurements would either need to be collected on separate days or the study restructured so all indirect measurements were obtained prior to direct measurements. Although simultaneous measurements would be possible for the radial arteries, comparisons would be confounded by body position (both arteries cannot be 'up' at the same time). Although a crossover design could be employed, 'white coat' induced by arterial catheterization would be anticipated to confound indirect SAP results taken after direct measurements. For all these reasons, direct SAP measurements were not collected.

Instead of using a crossover design, simultaneous measurement at both sites could have been considered to decrease any discordance due to moment by moment variation in blood pressure. This would have required either collection of all measurements with the foot elevated and the dog in a standing position, which is not a commonly used or recommended position for radial SAP measurement, or in lateral recumbency. Because the authors have found the coccygeal site to be particularly useful for accurate SAP measurement in dogs resistant to sitting or lying down, collection of coccygeal measurements in lateral recumbency was considered to limit the clinical applicability of the results. Finally, simultaneous cuff inflation at two sites is not consistent with standard measurement technique in clinical practice and it is unknown how that would have affected measurement-associated white-coat effect. The decision to maximize clinical relevance of the results by using a crossover design instead of collecting pressures simultaneously is unlikely to have substantively influenced the results given the lack of significant differences (statistically or clinically) among measurement results series (first measurement vs 2–6 vs 1–6) for either site and lack of period effects (which would have suggested a change in pressure in the time intervening between the two measurement series).

There were a number of additional limitations in the present study. In accordance with ACVIM guidelines, headphones were used for collection of SAP measurements in this study. The results, therefore, cannot be directly translated to other techniques including Doppler measurements taken without the use of headphones. A limited number of under-conditioned and sarcopenic dogs were available for enrollment in the study, with only 8 dogs in the BCS 1–2 group and only 6 dogs in the MCS 0–1 group. These demographics reflect the local population, which primarily includes pets of ideal or over-conditioned BCS. Although patients were recruited from both a healthy and an unhealthy population, most animals with higher BCS and MCS were not clinically ill. It is possible that diseases associated with hypertension, muscle wasting, and underconditioning–like chronic kidney disease–might affect accurate blood pressure measurement differently at one or both of the measurement sites. Six dogs in the study had hyperadrenocorticism (see Data S1), but there were few or no dogs enrolled that had other diseases, such as chronic kidney disease and diabetes mellitus, that commonly cause hypertension. Further study of all three factors together in a larger population of dogs with such disease processes is, therefore, recommended. Finally, a limited number of critically ill or hypotensive dogs were enrolled in this study, with only two dogs being classified as hypotensive. Because such animals often assume a laterally recumbent position, results for coccygeal artery measurements should be extrapolated to that patient population with caution.

## CONCLUSIONS

Indirect Doppler flow SAP measurements obtained at the radial and coccygeal artery sites were only moderately correlated. Mean radial measurements were 9 mmHg higher than coccygeal measurements, but discordance occurred in both directions. No association was found between measurements obtained at either site and age, weight, BCS, MCS, and anxiety score. Interestingly, no association was found between cuff size and blood pressure measurement at either site, suggesting that use of a cuff outside the recommended range might have little impact on the accuracy of results in a clinical setting.

Heart rate decreased throughout each data collection series, with results at the conclusion of SAP measurement significantly lower than those obtained at the beginning of SAP determination. However, there was no significant difference between the first SAP measurement, the mean of the additional five readings (BP 2–6), and the mean of all six readings (BP 1–6). In clinical situations where time is limited, the authors recommend that fewer blood pressure measurements be obtained instead of decreasing time devoted to patient acclimation to the environment and personnel.

### Funding

This work was supported by the University of Tennessee College of Veterinary Medicine Center of Excellence in Livestock Diseases and Human Health Summer Research Program. There was no external funding received for this study. The funders had no role in study design, data collection and analysis, decision to publish, or preparation of the manuscript.

### Grant Disclosures

The following grant information was disclosed by the authors:
University of Tennessee College of Veterinary Medicine Center of Excellence in Livestock Diseases.
Human Health Summer Research Program.

### Competing Interests

The authors have declare there are no competing interests.

### Author Contributions

- Allison P. Mooney conceived and designed the experiments, performed the experiments, analyzed the data, wrote the paper, prepared figures and/or tables, reviewed drafts of the paper.
- Dianne I. Mawby conceived and designed the experiments, performed the experiments, reviewed drafts of the paper.
- Joshua M. Price analyzed the data, contributed reagents/materials/analysis tools, wrote the paper, reviewed drafts of the paper.
- Jacqueline C. Whittemore conceived and designed the experiments, performed the experiments, analyzed the data, contributed reagents/materials/analysis tools, wrote the paper, prepared figures and/or tables, reviewed drafts of the paper.

## Animal Ethics

The following information was supplied relating to ethical approvals (i.e., approving body and any reference numbers):

The University of Tennessee Institutional Animal Care and Use Committee approved this project (protocol number 2426).

## Data Availability

The raw data has been supplied as a Supplementary File.

## Supplemental Information

Supplemental information for this article can be found online at http://dx.doi.org/10.7717/peerj.3101#supplemental-information.

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
