# Peer review of "Effects of various factors on Doppler flow ultrasonic radial and coccygeal artery systolic blood pressure measurements in privately-owned, conscious dogs"

_PeerJ, doi:10.7717/peerj.3101_

## Round 0.1 · original submission · Major Revisions

There is very serious concern about the study design that may preclude ultimate acceptance of this manuscript. These concerns include the absence of a "gold standard" measurement, which makes the value of either Doppler technique speculative. Other design issues that can lead to uncontrolled bias are explained in detail in the reviews. It will therefore be necessary to satisfactorily respond to all reviewer comments, and I will send your response back to at least one reviewer for re-review and a final recommendation concerning publication.

Reviewer 1 ·

Basic reporting

Manuscript is clear and well-written.

Experimental design

See general comments section.

Validity of the findings

See general comments section.

Additional comments

Abstract

Objective: Why did the investigators not include a “gold standard” measurement of blood pressure in their study? In other words, why not compare radial and coccygeal NIBP to direct ABP instead of each other?

Methods: Each investigator measures both BCS and MCS and thus is not blinded to the pressure value of the first measurement when performing the second blood pressure measurement. Why wouldn’t this lead to bias that favors agreement between the two measurement techniques? Why wasn’t this study conducted using one of the two following techniques instead:
1. One investigator listens for the Doppler sound and a second investigator reads the manometer
2. One investigator measures the coccygeal artery pressure and the other investigator measures the radial artery pressure without communicating the values to each other. The investigators alternate who is assigned the BCS vs. the MCS measurement.

Conclusions: The authors state “Initial measurement site can be based on patient and operator preference given lack of associations with patient variables…” It can be argued that we do not know whether either site is preferable because we have no gold standard comparison.

Methods:

The investigators are concerned about the effect of BCS on radial vs. coccygeal arterial pressure measurements and explain the importance of obtaining accurate pressures in hypertensive or hypotensive patients. However, the study population is comprised of healthy normotensive dogs. Is it possible that accuracy of site selection varies with blood pressure, and if so, could decisions to study only normal dogs affect their ability to evaluate whether one site is better than another in the clinical population of interest (hypertensive or hypotensive dogs)?

Was the sphygmomanometer calibrated? If so, how and when? If not, how do you know that the pressure measurements were accurate?

Why weren’t blood pressure measurements at the radial and coccygeal sites measured simultaneously? Blood pressure can potentially vary significantly in awake dogs from moment to moment. If the investigators were concerned that measurements could be different because of the blood pressure equipment used, the study could be designed to alternate each equipment set-up between the tail and arm and include this covariate in the analysis.

Discussion: Questions and concerns similar to those raised in Abstract and Methods sections.

·

Basic reporting

No comments

Experimental design

No comment

Validity of the findings

No comment

Additional comments

Review of the “Effects of various factors on Doppler flow ultrasonic
radial and coccygeal artery systolic blood pressure
measurements in privately-owned, conscious dogs”
(#14868)


General comment: The manuscript is well structured and the study is nicely carried out. My congratulations to the authors.


Specific comments:
Line 283 – Bodey instead of Body

Line 311 – No correlations to anxiety score should be attributed to low stress (shown in low heart rates) during the procedure

Was there a difference between tail and radial circumference %? That might have explained the difference in on average higher coccygeal BP.
It would be also more logical to expect higher BP in standing vs recumbent position, that means coccygeal would be higher, which was not the case.
Dogs and cats usually experience taking BP in front legs more annoying than on the tail and that what could have made the radial BP higher

Supplemental table BP database: I suggest adding measurement units to radial and tail circumference and BP.

---

## Round 0.2 · accepted · Accept

In production, please check the following footnote: "Riestar Direct" - should this be spelled "Riester Direct"?